# Role of Patient Sex in Early Recovery from Alcohol-Related Cognitive Impairment: Women Penalized

**DOI:** 10.3390/jcm8060790

**Published:** 2019-06-04

**Authors:** Amandine Luquiens, Benjamin Rolland, Stéphanie Pelletier, Régis Alarcon, Hélène Donnadieu-Rigole, Amine Benyamina, Bertrand Nalpas, Pascal Perney

**Affiliations:** 1Paul Brousse Hospital, APHP, 94800 Villejuif, France; amine.benyamina@aphp.fr; 2Paris-Saclay University, Univ. Paris-Sud, UVSQ, CESP, INSERM, 94800 Villejuif, France; pascal.perney@yahoo.fr; 3Service Universitaire d'Addictologie de Lyon (SUAL), CRNL, Université de Lyon, UCBL, INSERM U1028/CNRS 5292, CH Le Vinatier, F-69500 Bron, France; benjrolland@gmail.com; 4Addictions Department, Nîmes University Hospital, 30000 Nîmes, France; pelletiersteph@yahoo.fr (S.P.); regis.alarcon@chu-nimes.fr (R.A.); bertrand.nalpas@inserm.fr (B.N.); 5Addictions and Internal Medicine Department, Montpellier University Hospital, 34000 Montpellier, France; lndonnadieu@gmail.com; 6U 1058-mixte avec Université de Montpellier, Pathogenèse et contrôle des infections chroniques (PCCI), 34000 Montpellier, France

**Keywords:** cognitive recovery, alcohol-use disorder, gender, cognitive remediation, cognitive impairment, MOCA

## Abstract

Background: The objective was to explore the role of patient sex in cognitive recovery and to identify predictive factors for non-recovery in alcohol use disorder (AUD). Methods: All patients with AUD admitted to a residential addictions treatment center were systematically assessed at admission and after 6 weeks of abstinence in a controlled environment. The inclusion criteria were that patients were admitted for AUD with baseline alcohol-related cognitive impairment (baseline total Montreal Cognitive Assessment (MoCA) score < 26) and reassessed at 6 weeks (*n* = 395). A logistic regression model was built to determine the influence of sex on recovery status (MoCA < or ≥ 26) taking into account the interaction effect of sex with alcohol consumption on cognitive function. Results: The mean age was 50.10 years (SD = 9.79), and 27.41% were women. At baseline, the mean MoCA scores were 21.36 (SD = 3.04). Participants who did not achieve recovery (59.3% of women vs 53.8% of men) had lower total MoCA scores at baseline. The 2 factors that was significantly and independently associated with non-recovery and with a non-zero coefficient was being a woman and initial MoCA score (respective adjusted odds ratios (AOR) = 1.5 and 0.96, *p*-values < 0.05). Conclusions: These results could influence the time required in a controlled environment to maintain abstinence and the duration of in-care for women.

## 1. Introduction

### 1.1. Cognitive Impairment in Alcohol Use Disorder

Alcohol-related cognitive disorders are now well documented and recognized as being part of the alcohol use disorder (AUD) burden and a public health issue which need to be addressed, not only in terms of prevention, but also as potential therapeutic targets [1,2]. In addition to pre-existing cognitive vulnerability to addictive disorders [3] often involving less inhibitory capacities, specific deficits related to heavy drinking have been reported in AUD patients. The functions most affected by this toxicity are attention, working memory, inhibitory control including interference control, motor response inhibition [4], updating and shifting abilities [5], decision-making in social and uncertain situations [6], planning, visuospatial memory and problem-solving abilities [7,8]. Specific structural brain abnormalities and dysfunctional connectivity have been demonstrated in alcohol-dependant individuals [9,10,11,12,13]. The mechanisms responsible for these impairments are multiple and still being explored. Decreased levels of neurotrophin Brain-Derived Neurotrophic Factor (BDNF) have been reported in chronic alcohol users [14]. Additionally, AUD could be associated with demyelinization and axonal loss [7]. Irrespective of the mechanisms implied, cognitive impairment could be linked to a higher risk of relapse and low motivation to quit [15,16].

### 1.2. The Possibility of Cognitive Recovery

Most cognitive functions are recovered after alcohol withdrawal. However, the chronology and factors influencing the recovery process remain only partially explored. The various cognitive functions affected do not seem to recover in the same timeframes. Inhibition deficits appear to persist longer after detoxification than deficits in working memory. While spontaneous recovery has been demonstrated by the simple effect of abstinence, and therefore of stopping intoxication, cognitive remediation could speed up or enhance cognitive recovery [17]. Moreover, individual differences have been observed in the path to recovery [18]. The two factors most reported as impairing cognitive recovery are age [19,20] and the number of previous detoxifications due to reduced brain plasticity [21,22]. Additionally, other specific factors might influence brain recovery processes, i.e., genotype-dependent neuronal (re)growth, interfering effects of psychiatric comorbidities, additional smoking or use of marijuana, early onset of alcohol abuse, and sex-specific neural recovery effects [23]. It therefore appears crucial to document sex differences in the processes leading to cognitive impairment and subsequent recovery [24].

### 1.3. Role of Patient Sex in the Onset and Recovery of Alcohol-Related Cognitive Impairment

There is accumulating evidence of sex differences in pharmacokinetic responses to alcohol consumption and metabolism, for example in enzymatic activity between male and female patients for the enzymes that actively participate in ethanol oxidation in the liver [25]. It has been demonstrated that women are more vulnerable to alcohol than men. It is clear that for identical alcohol consumption, the morbidity and mortality risk is higher for women than for men [26]. The risk cannot be explained by only a difference in body size leading to higher blood concentrations for the same weight and a lower volume of distribution. The increased toxicity is multifactorial. It could be partly explained by lower gastric alcohol dehydrogenase activity in women and a suspected hormonal impact [27,28]. The female sex is also associated with increased inflammatory response to alcohol intoxication which may contribute to this increased toxicity [29]. Volumetric studies in human samples show that females may demonstrate increased volumetric brain loss with equal or lesser dependence histories than males, and different areas are impacted with women being especially prone to hippocampal damage [30]. Differences in neurotransmitter and endocrinological responses could influence neuronal function and viability, particularly during alcohol withdrawal [31]. Differences in the severity and nature of cognitive disorders [32] may be due to different mechanisms and could be explained in part by a paucity of trophic support and plasticity-related signaling in women compared to men [30]. However, other studies reported better recovery in women [33]. Recovery timeframes are also different for men and women. For instance, no change in short-term memory was observed at year 1 in abstinent from alcohol women in one study [34], while another demonstrated that recovery of white matter volume may occur sooner for women than for men [2]. Different impairments could have different timeframes and mechanisms with which sex could interfere.

The objective of this longitudinal study was to explore the role of sex in early cognitive recovery among abstinent in-patients with alcohol-related cognitive impairment and to identify the predictive factors for non-recovery taking into account the differential effect of alcohol consumption. Our hypothesis was that sex would be a significant factor associated with early recovery difficulties, taking into account a differential effect of alcohol on women.

## 2. Material and Methods

### 2.1. Population

All the patients admitted to the residential treatment unit of the University of Nîmes Addictions Center were systematically and routinely assessed at admission and after 6 weeks of abstinence in a controlled environment. Inclusion criteria were as follows: (1) DSM-5 criteria for AUD; (2) baseline alcohol-related cognitive impairment defined by a baseline total Montreal Cognitive Assessment (MoCA) score < 26 [35]; and (3) completion of the facility’s 6-week program including reassessment with a MoCA score at 6 weeks. The exclusion criteria were as follows: (1) Severe comorbid neurological or psychiatric disease (Alzheimer disease, psychosis, past medical history of stroke or coma); (2) infection by HIV; (3) difficulties with the French language. Usual care was provided during the in-between period, and patients participated in a cognitive remediation program with identical modalities for men and women. Cognitive remediation was undertaken by different professionals and particularly a neuropsychologist, an occupational therapist and a sports trainer. The program was based on mental and physical training targeting executive functions, visuospatial skills, attention, working memory, episodic memory, metacognition, and emotion regulation. Addictive comorbidities were not excluded but cannabis use disorder at baseline was recorded. All patients remained abstinent from alcohol, cannabis and other substances during the 6 weeks spent in this controlled environment.

### 2.2. Measures

The Montreal Cognitive Assessment (MoCA) test was designed as a rapid screening tool for mild cognitive dysfunction. It assesses different cognitive domains—attention and concentration, executive functions, memory, language, visuo-constructional skills, conceptual thinking, calculations, and orientation. Regarding all the screening tools for alcohol-related cognitive impairment, the Montreal Cognitive Assessment [36] has been rated comparatively as having the best psychometric qualities [37]. The maximum possible score is 30 points and the normal value ≥26 has recently been validated for patients with AUD [35].The <26 cut-off has been extensively used the later years in the addiction field, which is in addition to its extensive and older use in neurological contexts [15,38,39].Version 7.1 was used at baseline and version 7.2 of the tests was used for the re-assessment at 6 weeks to avoid memory biases. The instructions are the same in both versions, but all exercises use different values or words. The 7.2 version consisted of all elements of the 7.1 MoCA test and are replaced with equivalent elements that respect complexity, level of difficulty, administration and scoring time, linguistic frequency, cultural compatibility, cognitive domain specificity. For instance, animals to be named were lion, rhino, and camel in the 7.1 version and snake, elephant, and crocodile in the 7.2 version. Difference in scores observed between the 7.1 and 7.2 French MoCA versions in each subject cohort were not considered clinically significant in the validation study [40]. A proxy was used for recovery of a total MoCA score ≥26 at 6 weeks. A binary approach considering return to a MoCA score above the normal cut-off for recovery has been used before. A change in Moca in short periods have been reported before and considered to demonstrate a cognitive recovery [41]. A MoCA score has previously been used to classify subjects with cognitive recovery after cognitive training in the context of a stroke, using the very same cut-off of <26 [41]. Other variables of interest were systematically recorded, including parameters previously described in the literature as potential predictive factors of poorer recovery from alcohol-related cognitive impairment, i.e., sex, age [42], age of onset of AUD [43], number of previous detoxifications [21,23], daily alcohol consumption, cannabis use disorder [44] and tobacco use disorder [21], body mass index (BMI), metabolic syndrome [45], and cirrhosis [46].

### 2.3. Statistical Analysis

A descriptive analysis of the sample is provided according to cognitive recovery status and sex. Qualitative variables were converted into dummy variables. Between-group differences were explored with respect to collected variables after conversion into dummies using the Mann Whitney test. A logistic regression model was then built, with recovery status (i.e., 6th week MoCA score < or ≥ 26, respectively) as the dependent variable and all other previously mentioned parameters as independent variables, taking into account the interaction effect of sex and alcohol consumption, i.e., the effect of the combination of sex and alcohol consumption on cognitive function, and the reported corresponding adjusted coefficients. Missing data were imputed with the median method before logistic regression. All analyses were performed with Python in Anaconda.

## 3. Results

### 3.1. Description of the Whole Sample and Per Recovery Group

Sample characteristics are presented in Table 1 (*n* = 395) for the whole sample and per recovery group.

### 3.2. Description by Sex

The baseline total MoCA scores in men and women were respectively 21.34 (SD = 3.40) and 21.41 (SD = 3.05). It was the 4th mean detoxification for both the men and the women. The mean age was 50 in men and 52 in women. The age of onset of AUD was later in women than in men: 36 vs. 33 years. Respectively, 16 and 17% of men and women presented a comorbid cannabis use disorder, 73 and 70% a tobacco use disorder, 18% and 11% cirrhosis. The BMI was 26 in both the men and the women and 19% of patients of both sexes presented metabolic disorders.

### 3.3. Comparison of Recovered and Non-Recovered Patients after 6 Weeks of Abstinence in a Controlled Environment

No significant difference was found between recovery groups for any baseline variable except for the total MoCA score and some MoCA tasks—visuospatial, attention (3d task), language (1st task) and delayed recall (*p*-value < 0.05).

### 3.4. Change in MoCA Scores

The average change in total MoCA scores was 3.20 (SD = 3.48) for the whole sample. The total MoCA score at 6 weeks of abstinence in the controlled environment was 24.56 (SD = 3.30). Both the men and the women gained an average of 3 points (respectively 3.12 (SD = 3.63) and 3.43 (SD = 3.10)). Changes in MoCA scores compared to baseline values are shown in Figure 1. At 6 weeks, 45% of patients had recovered. Non recovery occurred in 59.3% of women vs. 53.8% of men.”

### 3.5. Predictive Factors of Non-Recovery at 6 Weeks

The only factor significantly and independently associated with non-recovery and with a non-zero coefficient was being a woman. The results from the logistic regression explaining non-recovery are presented in Table 2. The baseline total MoCA scores were significantly associated with recovery, but with very low coefficient, and the interaction effect of sex and alcohol consumption with a zero-coefficient (*p*-value < 0.05). The Adjusted Odds Ratio for non-recovery for women was 1.5 (*p*-value < 0.05). In other words, women had a 50% increased risk of not recovering from cognitive impairment at 6 weeks of abstinence compared to men.

## 4. Discussion

This study of a large sample of systematically included recently-detoxified participants with AUD having benefited from cognitive remediation shows sex-related differences in early-recovery at 6 weeks in a controlled environment, using return to a normal MoCA score as a proxy for recovery. It was found that being a woman was a risk factor for non-recovery from alcohol-related cognitive impairment. This finding has several implications. First, it highlights the importance of taking into account the interaction effect of sex and alcohol consumption, and of exploring the role of sex in the understanding of cognitive impairment and recovery with the perspective of implementing specific therapeutic strategies. Secondly, introducing this interaction effect to explore recovery led to non-replication in our study of previous findings regarding the role of nutritional status, cannabis and tobacco use, cirrhosis, and drinking/detoxification history. It should be noted that most previous studies explored cognitive improvement instead of recovery status. More precisely, it was found that improvement of particular cognitive functions or of global cognitive functioning can occur without a return to normal values as assessed with a cognitive battery or a screening test. This study was interested in exploring recovery, defined as the return to a normal value above the standard MoCA cut-off score, in a binary perspective.

Possibly, the influence of sex on recovery was insufficiently taken in consideration in previous studies, although it appears critical for a global understanding of non-recovery risk factors. Another interpretation would be that study limits prevented replicating these results; in particular, the samples were unbalanced for cirrhosis versus no cirrhosis, and for cannabis use disorder versus no cannabis use disorder. Further, exploring the role of these factors through matched designs among men and women could be interesting.

Nevertheless, in this particular population, the initial MoCA score explained a very little part of the variability of change in MoCA score at 6 weeks (significant factor with a very low β =−0.04, that shows us that for each point of initial MOCA score, the chance to recover decreases by 4%).Being a female increased the risk of non-recovery by 50%, all other things being equal, including the initial MoCA score. The tasks with lower initial scores in participants who did not reach recovery at 6 weeks had in common that they implied working memory and short-term memory. The results could reflect previous findings on delayed recovery in short term memory in women [34]. Based on these findings, it could be warranted to reinforce cognitive remediation targeting working memory in women.

Our results suggest that recovery timeframes are different in men and women and that in women, there could be a delay in the recovery of some cognitive functions rather than real recovery barriers, which could be due to different or co-existing mechanisms of impairment. Psychiatric comorbidities have been found to be associated with the detection of cognitive disorders using the MoCA test, particularly depression and agoraphobia [47]. As the high prevalence of depressive disorders in women with AUD [43] and other substance use disorders [48] is known, it would have been interesting to measure psychiatric comorbidities, and this is a limitation of our study. In addition, alcohol could interact with other factors known to vary depending on sex, such as resilience to stress demonstrated in female rats, and produce or worsen cognitive impairment in particular situations [49]. Similarly, alexithymia has been shown to be a possible mediating factor between depression and AUD, with its effect differing according to the sex of the subject [50]. Co-existing factors with opposite effects could make it difficult to interpret sex differences in recovery. In the future, psychological functioning measures could interestingly complete our findings to further explain their underlying mechanisms.

### Strength and Limits

The prospective and not cross-sectional design of our study is a strength and responds to a lack in current literature reporting on the evolution of cognitive impairment in alcohol-dependant patients [22]. Recovery status had not been confirmed by a comprehensive neuropsychological battery. The results remain to be confirmed by a study confirming the recovery status by a neuropsychological battery, as conducted in Aben et al. 2018 [51]. Moreover, the <26 cut-off is still debated in some populations, and particularly in the elderly [52,53]. Patients who withdrew from the therapeutic program and could not be re-assessed were not included. Moreover, our study is monocentric and the results should be reproduced in a multicentric design. Our sample is therefore not representative of all patients with alcohol-related cognitive impairments. The mean gain of MoCA was 3 points. It could be considered as a limited gain to demonstrate cognitive recovery. However, a smaller gain of 2.10 has previously been considered to be clinically significant by another team [41]. Logistic regression allows multiple explanatory variables being analysed simultaneously, meanwhile reducing the effect of confounding factors and analysing the role of covariates all things being equal (i.e., independently from the role of the other covariates, and here, in particular, the role of sex independently from the initial MoCA score and inversely). However, it cannot strictly replace a design where women and men would have been matched on the initial MoCA score, which would be an even more robust design. Women were fewer than men in our sample. Further studies including other possible factors of resistance to recovery, such as psychiatric co-morbidities, and in particular, depression and anxiety, psychological functioning, and biomarkers, could interestingly complete these results and assist with their interpretation. Irrespective of the underlying mechanisms, the increased risk of non-recovery in women after 6 weeks of abstinence and cognitive remediation is an issue that could justify sex-tailored strategies to guarantee a good prognosis and prevent relapse. These results could have implications on the time required spent in a controlled environment to maintain abstinence and on the duration of in-care for women to allow them time to reach cognitive recovery and recovery of the functions targeted by cognitive remediation.

## Figures and Tables

**Figure 1 jcm-08-00790-f001:**
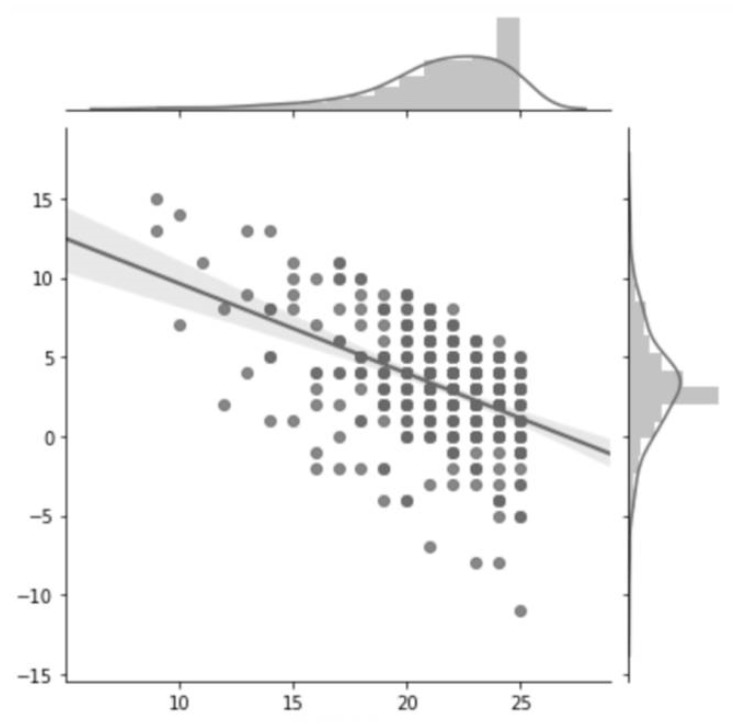
Change in total MOCA score at 6 weeks as a function of the initial MOCA score.

**Table 1 jcm-08-00790-t001:** Baseline characteristics of patients with alcohol-related cognitive impairment at admission (total Montreal Cognitive Assessment (MOCA) score < 26), and per recovery status after 6 weeks (total MOCA score ≥ or < 26).

	All	Sex		Recovery at Discharge	
	*N* = 395	Male *N* = 286	Female *N* = 108	Yes *N* = 177	No *N* = 218
Sex (female), *n* (%) (*n* = 394)	108 (27.34%)	-	-	44 (25%)	64 (29.36%)
Age, mean (SD) (*n* = 395)	50.10 (9.79)	49.56 (9.78)	51.56 (9.74)	50.18 (9.58)	50.03 (9.98)
Number of years of education, mean (SD) (*n* = 389)	10.57 (2.44)	10.55 (2.47)	10.60 (2.39)	10.71(2.35)	10.45 (2.51)
Number of previous detoxifications, mean (SD) (*n* = 252)	2.92 (2.71)	2.93 (2.93)	2.89 (2.13)	2.83(2.79)	3.00 (2.66)
Age at AUD onset, mean (SD) (*n* = 365)	33.84(11.43)	32.97 (11.92)	36.18 (9.75)	34.60(11.53)	33.22(11.34)
Daily alcohol consumption (g), mean (SD) ( *n* = 329)	211.45 (116.41)	225.27 (120.28)	171.10 (90.94)	215.14 (123.91)	208.58 (110.48)
Cannabis use disorder (yes), *n* (%) (*n*=395)	66 (16.71%)	46 (16.43%)	18 (16.67%)	30 (16.95%)	36 (16.51%)
Tobacco use disorder (yes), *n* (%) (*n*=395)	285 (72.15%)	208 (72.72%)	76 (70.37%)	127 (71.75%)	158 (72.48%)
BMI, mean (SD) (310)	26.03 (5.43)	25.91 (5.44)	26.33 (5.44)	26.26 (5.57)	5.32 (25.86)
Metabolic disorders, *n* (%) (*n*=394)	75 (18.99%)	55 (19.29%)	20 (18.51%)	29 (16.38%)	46 (22.02%)
Cirrhosis (yes), *n* (%) (*n* = 395)	65 (16.46%)	53 (18.53%)	12 (11.11%)	32 (18.08%)	33 (15.14%)
MOCA score at admission					
Total score, mean (SD) * (*n* = 395)	21.36 (3.04)	21.34 (3.04)	21.42 (3.05)	22.25 (2.30)	20.65 (3.36)
Visuospatial/executive (/5), mean (SD) *	2.83 (1.30)	2.88 (1.31)	2.69 (1.29)	2.10 (1.270)	2.71 (1.32)
Naming (/3), mean (SD)	2.90 (0.33)	2.90 (0.34)	2.90 (0.30)	2.91 (0.31)	2.89 (0.35)
Attention_1 (/2), mean (SD)	1.34 (0.67)	1.31 (0.68)	1.43 (0.64)	1.37 (0.65)	1.32 (0.69)
Attention_2 (/1), mean (SD)	0.87 (0.34)	0.89 (0.32)	0.81 (0.39)	0.89 (0.32)	0.36 (0.85)
Attention_3 (/3), mean (SD) *	2.27 (0.93)	2.36 (0.89)	2.03 (0.98)	2.45 (0.77)	2.12 (1.02)
Language_1 (/2), mean (SD) *	1.69 (0.53)	1.69 (0.44)	1.69 (0.57)	1.76 (0.48)	1.63 (0.57)
Language_2 (/1), mean (SD)	0.29 (0.45)	0.27 (0.44)	0.33 (0.47)	0.32 (0.47)	0.26 (0.44)
Abstraction (/2), mean (SD)	0.78 (0.68)	0.76 (0.69)	0.83 (0.68)	0.80 (0.70)	0.77 (0.67)
Delayed recall (/5), mean (SD) *	2.91 (1.36)	2.82 (1.41)	3.16 (1.19)	3.15 (1.29)	2.72 (1.39)
Orientation (/6), mean (SD) *	5.49 (0.72)	5.47 (0.71)	5.55 (0.74)	5.64 (0.58)	5.37 (0.80)

*: significant between groups with recovery status differences (Mann Whitney test).

**Table 2 jcm-08-00790-t002:** Predictive factors of non-recovery from cognitive impairment after 6 weeks of abstinence: Multivariate analysis (logistic regression).

	Coefficient	*t*	*p* > |*t*|	[0.025	0.975]
Intercept	1.49	5.23	0.000	0.932	2.054
Age	−0.00	−0.49	0.622	−0.008	0.005
Sex	0.38 *	3.06	0.002	0.138	0.630
Number of years of education	0.01	0.79	0.428	−0.013	0.030
Number of previous detoxifications	0.00	0.16	0.872	−0.020	0.024
Age at AUD onset	−0.00	−0.96	0.338	−0.008	0.003
Daily alcohol consumption	0.00	0.83	0.408	−0.000	0.001
Interaction effect of sex and daily alcohol consumption	−0.00 *	−2.87	0.004	−0.003	−0.001
Cannabis use disorder	0.00	0.06	0.953	−0.134	0.142
Tobacco use disorder	−0.01	−0.21	0.832	−0.131	0.105
BMI	0.00	0.08	0.934	−0.010	0.011
Metabolic disorders	0.08	1.26	0.210	−0.045	0.206
Cirrhosis	−0.06	−0.83	0.405	−0.192	0.078
Total MOCA score at baseline	−0.04 *	−5.21	0.000	−0.061	−0.028

* significant factor independently associated with recovery.

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
