# Peer review of "Role of Patient Sex in Early Recovery from Alcohol-Related Cognitive Impairment: Women Penalized"

_jcm, 2019, doi:10.3390/jcm8060790_

Reviewer 1 Report

This manuscript reports on a study of the cognitive “recovery” of individuals with DSM-5 alcohol use disorder from before to after a 6 week abstinence in one specific treatment facility.  The sample of male and female adults with a mean age in their 50s had a mean of 4 previous detoxifications. Mean Montreal Cognitive Assessment scores were in the impaired range before abstinence.  The authors treat a score of 26 or more as recovered.  Logistic regression was used to test the importance of demographic and other characteristics to being “recovered” at 6 weeks.

While the question, methods and findings discussed in this manuscript are interesting, and some what unique, they are limited  by the sample, measure of recovery and treatment of the MoCA scores.

In the abstract, the reviewers note that a score of 26 or above is normal.  They do not review any literature that used such a score to connote “recovery” nor do they discuss any work that has normed the expected or possible change in MoCA score after 6 weeks.  They do mention the use of a different version of the MoCA at 6 weeks, but provide no citations to help the reader understand if this change is enough to reduce the re-testing bias on the score.  Work on the use of 26 as the norm for the MoCA is ongoing; it would have been important to show that this norm was correct for this population (outside of one citation to another place). The lack of sufficient sophistication in the literature review complicates acceptance of the measure of the MoCA score of 26 and above as an accurate measure for cognitive recovery, especially in this sample among whom the majority have had repeated detoxifications.

The measures section is adequate except for concerns related to taking a score for “normal” and considering this an appropriate marker for “cognitive recovery.”  The analysis section was sufficient.

Results were briefly reported. Tables were interpretable.

The discussion would have been strengthened by discussing the “non-zero coefficient” part of the analysis. Baseline MoCAs between the two groups were significantly different.  Given that “recovery” the way it was defined was about 3 points on the MoCA, and given that the two groups varied at baseline by about 2 points, this difference seemed clinically (by your definition) significant, IF the outcome itself was clinically significant.  This too is a concern. I do not think that most people will consider this change to show what should be considered recovery.  Other research has shown that some people who are not impaired using a gold standard neurological assessment actually score lower on the MoCA than a 26. See, for example   Carson N,  Leach L, Murphy KM. A re‐examination of Montreal Cognitive Assessment (MoCA) cutoff scores. International Journal of Geriatric Psychiatry. 2018; 33(2): 379-388.  Given these limitations, and given that a convenience sample from one facility, the findings are overinterpreted.

Author Response

We are very grateful to the editor and the reviewers for their time and for giving us the opportunity to improve our manuscript thanks to their relevant comments. Please find below our responses to the reviewers’ and editor’s comments in red, and attached the 2nd  version of the manuscript.

Reviewer 1

This manuscript reports on a study of the cognitive “recovery” of individuals with DSM-5 alcohol use disorder from before to after a 6 week abstinence in one specific treatment facility.  The sample of male and female adults with a mean age in their 50s had a mean of 4 previous detoxifications. Mean Montreal Cognitive Assessment scores were in the impaired range before abstinence.  The authors treat a score of 26 or more as recovered.  Logistic regression was used to test the importance of demographic and other characteristics to being “recovered” at 6 weeks.

While the question, methods and findings discussed in this manuscript are interesting, and some what unique, they are limited  by the sample, measure of recovery and treatment of the MoCA scores.

(1) In the abstract, the reviewers note that a score of 26 or above is normal.  They do not review any literature that used such a score to connote “recovery” nor do they discuss any work that has normed the expected or possible change in MoCA score after 6 weeks.  (2)  They do mention the use of a different version of the MoCA at 6 weeks, but provide no citations to help the reader understand if this change is enough to reduce the re-testing bias on the score.  (3) Work on the use of 26 as the norm for the MoCA is ongoing; it would have been important to show that this norm was correct for this population (outside of one citation to another place). The lack of sufficient sophistication in the literature review complicates acceptance of the measure of the MoCA score of 26 and above as an accurate measure for cognitive recovery, especially in this sample among whom the majority have had repeated detoxifications.

R1. A binary approach considering return to a MoCA score above a cut-off for recovery has been used before in stroke field.  Change in Moca in short periods have been reported before and considered to demonstrate a cognitive recovery (see for instance 8-week change in stroke in  Zhao, 2019, We added the following justification of our approach: “A binary approach considering return to a MoCA score above the normal cut-off for recovery has been used before.  Change in Moca in short periods have been reported before and considered to demonstrate a cognitive recovery (Zhao, 2019). “

However, to take into account the reviewer’s comment, we reformulated the methods and discussion section to highlight the fact that we report a proxy for recovery in the lack of more detailed information available. We also added that results remain to be confirmed by a study confirming the recovery status by a neuropsychological battery:

Methods: P3l116: “We used as a proxy for recovery a total MoCA score ≥26 at 6 weeks.

Discussion: p4l175 “, using return to a normal MoCA score as a proxy for recovery”

P7l219“ Recovery status had not been confirmed by a comprehensive neuropsychological battery. Results remain to be confirmed by a study confirming the recovery status by a neuropsychological battery, as conducted in Aben et al., 2018”

R2 We thank the reviewer for this comment and giving us the opportunity to illustrate the difference between the 2 versions of the MoCA used at the 2 timepoints : “The instructions are the same in both versions, but all exercises use different values or words. In 7.2 version consisted all elements of the 7.1 MoCA test are replaced with equivalent elements that respect complexity, level of difficulty, administration and scoring time, linguistic frequency, cultural compatibility, cognitive domain specificity. For instance, animals to be named were: lion, rhino, and camel in the 7.1 version and snake, elephant, and crocodile in the 7.2 version. Difference in scores observed between the 7.1 and 7.2 French MoCA versions in each subject cohort were not considered clinically significant in the validation study (Nasreddine, 2016).”

R3 The<26 cut-off use rely on an exponential literature having used it in the context of addiction with very similar patients. We justified that point in the discussion.

P3l115 “The<26 cut-off has been extensively used the later years in the addiction field, which is in addition to its extensive and older use in neurological contexts (Viswam, 2018, Gautron, 2018 Ridley, 2018)” .

However, We added in the limit section the point that the<26 cut-off is still debated in some populations to take into account the reviewer’s comment P7l222. :” Moreover, the<26 cut-off is still debated in some populations, and particularly in the elderly (Thomann, 2018, Carson, 2018)(”

 (4) The measures section is adequate except for concerns related to taking a score for “normal” and considering this an appropriate marker for “cognitive recovery.”  The analysis section was sufficient.

Results were briefly reported. Tables were interpretable.

R4 We thank the reviewer for this comment.

 (5) The discussion would have been strengthened by discussing the “non-zero coefficient” part of the analysis. Baseline MoCAs between the two groups were significantly different.  Given that “recovery” the way it was defined was about 3 points on the MoCA, and given that the two groups varied at baseline by about 2 points, this difference seemed clinically (by your definition) significant, IF the outcome itself was clinically significant.  This too is a concern.

R5 The points highlighted by the reviewers are correct: groups of recovery have more exactly1.6 points of difference in initial MoCA score.  We modified the sentences with the zero-coefficient”, to explain further the respective importance of the covariates in the model and not to neglect the independent role of initial MoCA:

P1l27 Abstract: “The 2 factors that was significantly and independently associated with non-recovery and with a non-zero coefficient was being a woman and initial MoCA score (respective AOR = 1.5 and 0.96, p-values<0.05).”< span="">

P6 l177 Results: “Baseline total MoCA scores were significantly associated with recovery, but with very low coefficient, and the interaction effect of sex and alcohol consumption with a zero-coefficient (p-value<0.05). The Adjusted Odds Ratio for non-recovery for women was 1.5 (p-value<0.05).”< span="">

P7l 207 Discussion: “Nevertheless, in this particular population, the initial MoCA score explained a very little part of the variability of change in MoCA score at 6 weeks (significant factor with a very low β =-0.04, that shows us that for each point of initial MOCA score, the chance to recover decreases by 4%).”

However, we want to highlight the fact that logistic regression allows determining the role of each covariate independently from the others through adjustment.  In the logit model the log odds of the outcome is modeled as a linear combination of the predictor variables: logit[π (X)] = β0 + β1X1 + β2X2 + : : : + βpXp. Logistic regression allows multiple explanatory variables being analyzed simultaneously, meanwhile reducing the effect of confounding factors. The coefficient β1 is such that e β 1 is the odds ratio for a unit change in X, and in general, for a change of z units, the OR= e 1  = (e β 1) z. Here, the multivariate model adjusted model allows analyzing the role of covariates all things being equal (i.e. independently from the role of the other covariates, and here in particular the role of sex independently from the initial MoCA score and inversely. The exp(β) of a continuous variable represents the increment of the chance of an event related to each unit increment on the explanatory variable (Sperandei, 2014). As regarding the initial MoCA score,  exp(-0.04) = 0.96 , it shows us that for each point of initial MOCA score, the chance to recover decreases by 4%. The between group difference is then not an issue. However, it of course can’t strictly replace a design where women and men would have been matched on the initial MoCA score, which would be an even more robust design.   We added that point in the limit section:

P8l241 “Logistic regression allows multiple explanatory variables being analyzed simultaneously, meanwhile reducing the effect of confounding factors and analyzing the role of covariates all things being equal (i.e. independently from the role of the other covariates, and here in particular the role of sex independently from the initial MoCA score and inversely). However, it can’t strictly replace a design where women and men would have been matched on the initial MoCA score, which would be an even more robust design.”

 (6)  I do not think that most people will consider this change to show what should be considered recovery. 

R6 We added the following paragraph to justify our position and take into account the reviewer’s comment:

p8l229 “The mean gain of MoCA was of 3 points. It could be considered as a limited gain to demonstrate cognitive recovery. However, a smaller gain of 2.10 has previously been considered to be clinically significant by another team (Zhao, 2018).”

(7) Other research has shown that some people who are not impaired using a gold standard neurological assessment actually score lower on the MoCA than a 26. See, for example   Carson N,  Leach L, Murphy KM. A re‐examination of Montreal Cognitive Assessment (MoCA) cutoff scores. International Journal of Geriatric Psychiatry. 2018; 33(2): 379-388.  

R7 See R1

 (8) Given these limitations, and given that a convenience sample from one facility, the findings are overinterpreted.

R8 We thank the reviewer for this comment and added the monocentric nature of our design as a limit in the limit section:

P8l223 « Moreover, our study is monocentric and results should be reproduced in a multicentric design.”

For the other points please see previous responses

Reviewer 2 Report

The paper titled ‘Role of patient sex in early recovery from alcohol-related cognitive impairment: women penalized’ by Luquiens et al., present alcohol-related cognitive impairment considering gender. 

I have some questions to be explained.

 I think that the Table 1 should present data for woman and man separately. The population  of woman is significantly lower than man  (27.34%).

 Statistical analysis

It would be interesting to know how many missing data were simulated with the median method. 
ResultsPage 4, line 4, ‘Change in MoCA scores’ paragraph: ‘Among the non-recovered patients, 59.3% were women vs 53.8% of men’. If I understand correctly, the sum exceeds 100%.

Author Response

We are very grateful to the editor and the reviewers for their time and for giving us the opportunity to improve our manuscript thanks to their relevant comments. Please find below our responses to the reviewers’ and editor’s comments in red, and attached the 2nd  version of the manuscript.

Reviewer 2

The paper titled ‘Role of patient sex in early recovery from alcohol-related cognitive impairment: women penalized’ by Luquiens et al., present alcohol-related cognitive impairment considering gender. 

I have some questions to be explained.

(1) I think that the Table 1 should present data for woman and man separately. The population  of woman is significantly lower than man  (27.34%).

R1We added data for women and men in table 1 as suggested by the reviewer. We added in the limit section the fact that that there were fewer women than men in our sample:

P8l241 “Women were fewer than men in our sample.”

 (2) Statistical analysis

It would be interesting to know how many missing data were simulated with the median method. 

R2 We added all n= per variable in table 1 before median imputation.

(3)Results Page 4, line 4, ‘Change in MoCA scores’ paragraph: ‘Among the non-recovered patients, 59.3% were women vs 53.8% of men’. If I understand correctly, the sum exceeds 100%.

R3 We’re sorry for having been unclear. We corrected the sentence as follows

P6 l 165: “Non recovery occurred in 59.3% of women vs 53.8% of men.”

Round  2

Reviewer 1 Report

Thank you for making these changes to the manuscript.